# Potential impact of introducing vaccines against COVID-19 under supply and uptake constraints in France: A modelling study

Laurent Coudeville[1]*, Ombeline Jollivet[1], Cedric Mahé[1], Sandra Chaves[1], Gabriela B. Gomez[1,2]

1 Modelling, Epidemiology and Data Science, Sanofi Pasteur, Lyon, France, 2 Department of Global Health and Development, London School of Hygiene and Tropical Medicine, London, United Kingdom

* Laurent.Coudeville@sanofi.com

## Abstract

### Background

The accelerated vaccine development in response to the COVID-19 pandemic should lead to a vaccine being available early 2021, albeit in limited supply and possibly without full vaccine acceptance. We assessed the short-term impact of a COVID-19 immunization program with varying constraints on population health and non-pharmaceutical interventions (NPIs) needs.

### Methods

A SARS-CoV-2 transmission model was calibrated to French epidemiological data. We defined several vaccine implementation scenarios starting in January 2021 based on timing of discontinuation of NPIs, supply and uptake constraints, and their relaxation. We assessed the number of COVID-19 hospitalizations averted, the need for and number of days with NPIs in place over the 2021–2022 period.

### Results

An immunisation program under constraints could reduce the burden of COVID-19 hospitalizations by 9–40% if the vaccine prevents against infections. Relaxation of constraints not only reduces further COVID-19 hospitalizations (30–39% incremental reduction), it also allows for NPIs to be discontinued post-2021 (0 days with NPIs in 2022 versus 11 to 125 days for vaccination programs under constraints and 327 in the absence of vaccination).

### Conclusion

For 2021, COVID-19 control is expected to rely on a combination of NPIs and the outcome of early immunisation programs. The ability to overcome supply and uptake constraints will help prevent the need for further NPIs post-2021. As the programs expand, efficiency assessments will be needed to ensure optimisation of control policies post-emergency use.

**Data Availability Statement:** The model code used is available at GitLab: https://gitlab.com/SPMEGModels/covid-model.

**Funding:** Sanofi provided financial support to authors in the form of salaries but did not have any additional role in the study design, data collection and analysis, decision to publish, or preparation of the manuscript. The specific roles of the authors are articulated in the 'Contributors' section of the manuscript.

**Competing interests:** All authors are Sanofi employees and may hold shares and/or stock options in the company. This does not alter the adherence of the authors to PLOS ONE policies on sharing data and materials.

## Introduction

With the risk of continuous transmission of SARS-CoV-2 and the disruption of global economy, expectations and investments in research and development of vaccines to control the COVID-19 pandemic have been unprecedented. In just over a year, almost 100 candidates have started clinical testing, with 20 ongoing phase III trials globally [1, 2]. Of these, four vaccines have been approved for use after showing critical efficacy results [3–6]. Regulatory bodies have adapted approval processes, which in turn have shortened timelines from months to weeks, as shown by the fast-track reviews of the European Medicines Agency and the Federal Drug Administration in the USA [7, 8]. Under these accelerated timelines, vaccines have become available late 2020, with implementation starting end 2020 beginning 2021 [9]. However, in the presence of global demand, even with scaled-up production taking place before trials completion, supply is likely to be constrained in this first year of implementation.

In parallel to clinical trials, policy makers' efforts have been geared towards planning immunisation programs to maximise societal benefits and achieve an equitable distribution of limited vaccines. Previous modelling studies have explored the potential population impact of immunisation programs depending on vaccines and programs characteristics, such as efficacy, coverage, duration of protection elucidated by vaccines or whether these are effective in preventing symptomatic disease alone or preventing infection as well [10–14]. Authors have shown that a vaccine will need to be highly effective and the immunisation program will need to achieve high coverage to be able to obviate the need of non-pharmaceutical interventions (NPIs) to control the pandemic in the short term. Other studies investigated prioritisation strategies to optimise immunisation impact considering limited coverage [15, 16], including recent modelling by the Imperial College COVID-19 Response Team. In this study, the authors showed that the optimal allocation strategy (i.e. groups to prioritise and relative coverage achieved) within country will likely depend on the level of supply constraint of vaccines being introduced in 2021 [17].

In addition, vaccine hesitancy limiting the impact of an eagerly awaited immunisation program should not be underestimated [18, 19]. Surveys to date have shown a changing level of willingness to vaccinate as individuals' perceptions of risk have evolved [19–21]. Populations have become less willing to vaccinate with first-come vaccines and may be increasingly more willing to wait for additional data on safety and effectiveness in real life conditions [22]. Immunisation programs will likely see successive candidates [23] becoming available during 2021, and later vaccine entrants may play a role improving supply and, depending on further data available, possible uptake.

Here, rather than focusing on optimal conditions for a vaccination program to be successful, we aimed at identifying plausible scenarios for the short to mid-term impact of an immunisation program in France, considering the uncertain vaccine profile and likely variation of supply and uptake.

## Methods

### Model structure and calibration

We built on a previously published age-stratified compartmental transmission model of SARS-CoV-2 to examine the short-term impact of an immunisation program starting January 2021 in France [24]. Briefly, we expanded a standard Susceptible-Exposed-Infectious-Recovered (SEIR) structure to account for seasonality of SARS-CoV-2 transmission, levels of disease severity, and possibility of reinfection with reduced level of severity compared to the primary infection. Reinfections are being defined as any infection to Covid-19 following a period with

naturally-acquired or vaccine-induced immunity. This immunity period is assumed to have a median duration of one year in the base case. Future vaccination is modelled through dedicated compartments where duration of immunity can be modulated.

The model was calibrated using least-squares minimization to French surveillance data up to November 21, 2020. Three outcomes derived from two sources were used for calibration: number of symptomatic cases and deaths reported by the European Centre for Disease Prevention and Control and hospital admissions reported by Santé Publique France [25, 26]. Natural history parameters for SARS-CoV-2 infection were based on the U.S. Centers for Disease Control and Prevention best estimates [27], while parameters such as infection fatality ratio, hospitalization rates, and social contact matrices by age were locally sourced [28, 29]. Our base case scenario corresponds to a situation with moderate seasonality (20% amplitude in COVID-19 transmission with a peak transmission in January) and limited symptoms in case of reinfection (90% reduced symptom severity compared to primary infections). A full description of the model, parameterization, and calibration method is presented in S1 Text in S1 File.

## Non Pharmaceutical Interventions (NPIs)

NPIs have been widely implemented across the globe with varying levels of stringency and compliance. As we enter the second year of the pandemic, countries are balancing a changing epidemiology of re-emerging cases with population signs of fatigue to adhere to mitigation measures. In this analysis, we do not explicitly model the effect of discrete NPIs but assume a reduction of the effective reproduction number due to a change in measures in place (e.g., social distancing, curfew, lockdown, contract tracing) triggered by a predefined threshold. This threshold, as well as the targeted effective reproduction number and timing of relaxation, differs between the three levels of NPI response we considered (see S2 Table in S1 File). Two of these scenario responses are based on the evolution of the incidence of hospitalization rates observed in France. The first and second wave thresholds are defined by the peak number of hospitalizations observed in the first and second waves. These thresholds are 100 and 70 hospitalizations per million population per day, respectively. The latter was used as our reference case. The third scenario is based on a hypothesized government response that could tolerate high level of disease rates and be triggered by a higher threshold of 200 hospitalizations per million population per day. In addition to the threshold-based response, we also accounted for a relative reduction of exposure to infection in the vulnerable population (elderly and people with comorbidities) compared to the rest of the population due to a better adherence of this part of the population to distancing measures. The relative reduction was derived from the calibration in our reference case at approximately 30% (more information in S1 Text in S1 File). We considered that the vulnerable population no longer maintain a lower exposure to infection compared to the rest of the population when the roll-out of the vaccination program is completed towards the end of 2021.

## Vaccine profiles

Vaccines efficacy in clinical trials published to date has been measured for prevention of moderate to severe disease [3–6]. However, in addition to the reduction of symptomatic (mild, moderate and severe) disease, these vaccines could also protect against infection (and therefore preventing also asymptomatic disease and transmission). We considered both cases as there is evidence from the trials of prevention of disease and emerging (less definitive) real-world evidence of prevention of infection [30, 31]. In both cases, we considered that onset of protection starts one month after the administration of the first dose. In our reference scenario, a vaccine

protecting against infection with an efficacy varying from 50% to 90% for individuals without prior exposure to COVID-19 and a median duration of protection of one year [32].

## Immunisation programs

We characterised the government-driven immunisation programs across three dimensions: 1) implementation, 2) population's willingness to be vaccinated, and 3) supply of vaccines. For the implementation of the program, we assumed the scale-up of such program to follow the French COVID-19 scientific and vaccine committees' proposal and distributes the vaccine allotment into priority access categories [33]. High priority groups (i.e. HCW and people with professional risks) are prioritised in the first two months of the immunisation program, vulnerable adults (i.e. those with comorbidities and the elderly) follow in the roll-out over the next four months. After these groups are covered, the national program reaches other adults (i.e. 20–59 years without comorbidities). This stepwise scale up aims to consider other implementation constraints related to health system limitations such as availability of human resources, consumables among others.

With regards to uptake, an important constraint for a COVID-19 immunisation program pertains to the willingness of the population to get vaccinated. The uptake constraint, defined as the maximum achievable coverage with available doses for the whole target group, was set at 60% of any priority access group based on the surveys of willingness to vaccinate available for the French population. Namely, an early survey in France (end March 2020) showed a significant proportion (25%) of respondents being reluctant to accept vaccination [19]. This proportion increased to more than 30% of respondents late in April 2020 and to 40% by early August 2020 [21, 34]. We also considered scenarios where uptake constraints are relaxed from mid-2021, leading up to a maximum of 90% coverage in the adult population. In those scenarios, the relaxation is hypothesised to result from public experience in the use of vaccines in real-life conditions, further safety data becoming available, and/or additional vaccines approved with different efficacy and safety profiles.

Supply constraints are likely to play a role in 2021. To date, the European Commission has concluded contracts with companies covering a broad portfolio of vaccine candidates [35–40]. As communicated by the European Commission, allocation between countries will be on a population pro-rata distribution key [41]. However, not all vaccine candidates are expected to be successfully registered. We assumed in this study that one or more vaccines will be available during the first quarter 2021 (currently three vaccines have been approved by EMA). We also assumed that new supplies will become available mid-year 2021, either due to industrial scale-up of first vaccines or subsequent vaccines becoming available [42, 43]. Four supply constraint scenarios were defined depending on the number of doses available during the first and second half of 2021 (see S3 Table in S1 File). These scenarios aim to represent a range of supply possibilities, with a 'strong supply constraint' scenario assuming just under 18 million people could be vaccinated by end of 2021, and a 'weak supply constraint' scenario that allows the program to reach up to 37 million people by the end of 2021. As with uptake constraints, we considered additional scenarios where all constraints are relaxed from mid-2021 (due to availability of vaccine doses), which we called "relaxed weak supply and uptake constraint" and "relaxed strong supply and uptake constraints", moving away from limitations set at the beginning of the roll-out of the vaccination program.

## Analysis

Health benefits associated with the immunisation program were quantified as the reduction in COVID-19 events and events averted per 1000 vaccinations. We focused on hospitalizations

averted in the main text because this is a locally relevant indicator to assess health system stress which guided policy decisions. However, it is not the only indicator available and we present results for symptomatic cases and deaths averted in S1 Text in S1 File. We also assessed the impact of vaccination on the number of days with NPIs in place. Three timeframes were considered in our analysis (2021, 2022, and 2021–2022) while we discuss the possible evolution of COVID-19 beyond 2022. In our impact analysis, we include an uncertainty range of main outcomes reflecting a variation in vaccine efficacy from 50 to 90%.

The analysis is structured as follows. First, we looked at the progression of the COVID-19 epidemic in the absence of vaccines for varying health policy strategies policies, i.e. here with various thresholds based on hospitalization rates for NPI initiation. In these scenarios, NPIs are maintained throughout 2021 and 2022. In S1 File, we considered alternative timing for NPIs discontinuation (S4 Table in S1 File). Secondly, we present the impact of implementation scenarios compared to the no vaccine counterfactual. This counterfactual use, as a reference, the second wave threshold for NPIs initiation maintained throughout 2021 and 2022. It represents a conservative assumption with regards to vaccine impact.

Thirdly, we explored five vaccination scenarios: one uptake constraint scenario, two scenarios with limited supply (strong supply constraint, weak supply constraint) and two scenarios relaxing these supply and uptake constraints during the second semester of 2021 (relaxed strong supply and uptake constraint, relaxed weak supply and uptake constraint). Description of the scenario definition and the coverages achieved in these scenarios is presented in Table 1. For those scenarios including an increase in the supply and uptake of vaccines in the second semester of 2021, we assessed their incremental benefits compared to scenarios without this increase.

Finally, sensitivity analyses were performed on potential drivers of SARS-CoV-2 transmission (seasonality, severity of reinfection), level of public health response, and vaccine profile (vaccine that only reduces symptomatic disease instead of preventing infection).

**Table 1. Vaccine coverage by June 2021 and December 2021 for three groups prioritised.**

| Scenario[1] | Coverage by end of June 2021 | | | | Coverage by end of December 2021 | | | |
|---|---|---|---|---|---|---|---|---|
| | High priority[2] | Vulnerable adults[3] | Other adults | All Adults | High priority[2] | Vulnerable adults[3] | Other adults | All Adults |
| No vaccine | 0% | 0% | 0% | **0%** | 0% | 0% | 0% | **0%** |
| Uptake constraint | 60% | 60% | 0% | **33%** | 60% | 60% | 60% | **60%** |
| Strong supply constraint | 60% | 9% | 0% | **12%** | 60% | 38% | 0% | **24%** |
| Weak supply constraint | 60% | 52% | 0% | **30%** | 60% | 60% | 26% | **45%** |
| Relaxed strong supply and uptake constraint | 60% | 9% | 0% | **12%** | 90% | 90% | 42% | **68%** |
| Relaxed weak supply and uptake constraint | 60% | 52% | 0% | **30%** | 90% | 90% | 88% | **89%** |

[1]Scenarios defined: **Uptake constraint** reflects limited coverage rate due to a low willingness to be vaccinated in the population. **Strong supply constraint** reflects a limited amount of vaccines doses made available to the national program. In this case, the program is severely limited. **Weak supply constraint** reflects a limited amount of vaccines doses made available to the national program, in this case the program is moderately limited. **Relaxed strong supply and uptake constraints**: in this scenario, while the program is severely limited during the first half of the year, vaccine supply is increased during the second half of the year (higher production or new vaccines availability) and the public is more likely to be willing to vaccinate as the program has been in place for half a year. Therefore, we 'eased' the limitations of the program in the second semester. **Relaxed weak supply and uptake constraints**: in this scenario, while the program is moderately limited during the first half of the year, vaccine supply is increased during the second half of the year and the public is even more likely to be willing to vaccinate as the program has been in place for half a year. Therefore, we 'eased' the limitations of the program to achieve maximum coverage.

[2]High priority includes healthcare workers and professions at risk.

[3]vulnerable adults include elderly and adults with comorbidities.

## Results

We modelled the evolution of the COVID-19 pandemic in France after fitting to observed data for reported deaths, hospitalizations, and symptomatic cases (S1 Fig in S1 File). In Fig 1, we present hospitalization incidence in the absence of vaccination for varying levels of NPIs response maintained until the end of 2022. The level of NPI response defines not only the magnitude of peak incidence but also the number of activation periods for the 2021–2022 period. This number of activation periods ranges from four to seven as evidenced by the number of times incidence exceeds the predefined threshold. It is also noteworthy than incidence remains at a high level at the end of 2022 indicating the possibility for waves further to 2022.

Discontinuing NPIs during the scale-up of an immunisation program could lead to uncontrolled COVID-19 outbreaks. Stopping NPIs in January 2021 could lead to a peak incidence of hospitalizations seven times the peak incidence of the first wave. The impact of all vaccination scenarios would also be minimal in this case (from 5 to 12% over the 2021–2022 period), as individuals could be infected before having the opportunity to benefit from vaccine protection (S4 Table in S1 File).

### Potential immunisation program impact under uptake and supply constraints

Vaccination programs with uptake and supply constraints are not expected to allow interrupting NPIs in the short term (Fig 2). In the absence of vaccination, it is expected that NPIs would be in place for most of 2021 (327 days), this number reduces respectively to 297 [268–308], 203

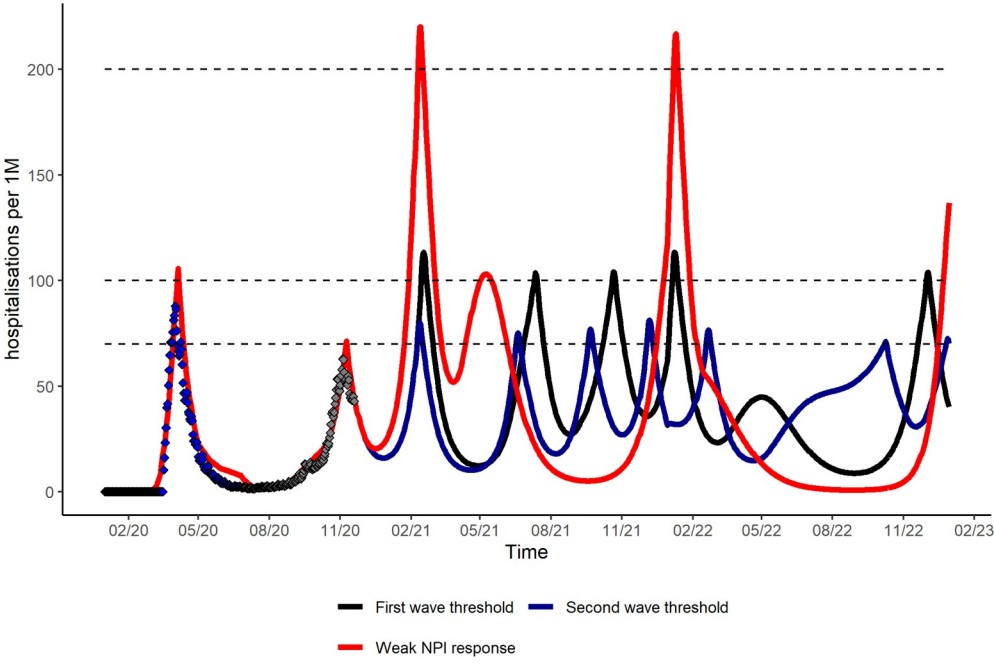

**Fig 1. Daily hospitalization incidence (rate per million population per day) in the absence of vaccination for varying level of public policy response, 2020–2022.** Based on reference scenario for seasonal variation in COVID-19 transmission (20%), severity of reinfection (90% less than primary infection), median duration of natural/vaccine immunity of one year and NPIs maintained until end of 2022. Black dotted lines correspond to the predefined hospitalization threshold of public health response. The black curve follows a scenario where the threshold for NPIs is based on the first wave (100 hospitalization per one million population). The blue curve represents a scenario where the threshold for NPIs is based on the second wave (70 hospitalizations per one million population). The red curve represents a scenario where the response is weak and the NPI threshold high (200 hospitalization per one million population).

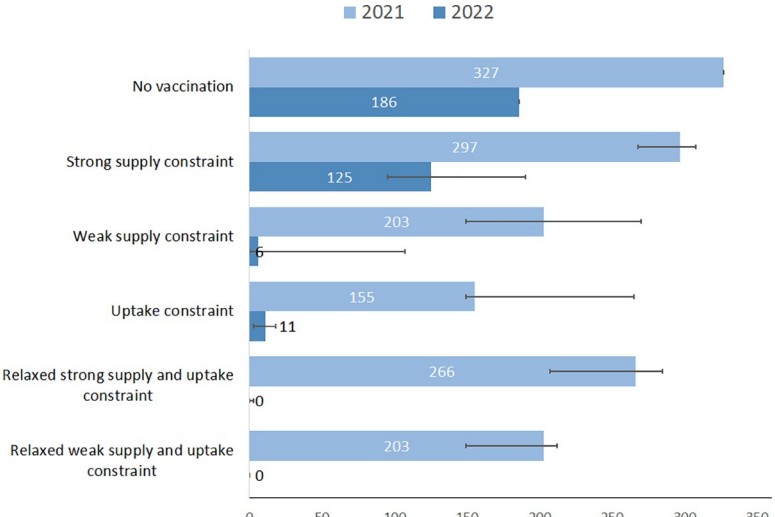

**Fig 2. Number of NPI activation periods in 2021–2022 and peak hospitalization incidence in the absence of NPIs.**
Based on reference scenario for disease characteristics, NPI response (second wave threshold), and vaccine profile
(protection against infection). The number of days and error bars corresponds respectively to reference, minimum and
maximum values for an efficacy of 70% ranging from 50% to 90%.

[149–270] and 155 [149–265] for the strong supply constraint, weak supply constraint and
uptake constraint scenarios. The situation improves in 2022, notably for the uptake constraint
scenario where we observe 11 [3–18] days with NPIs in place compared to 186 for the no vac-
cine counterfactual.

In Table 2, we assessed the reduction in hospitalizations due to the immunisation programs
compared to a no vaccine counterfactual assuming NPIs are discontinued after immunisation

**Table 2. Cumulative incidence of COVID-19 hospitalization per million population and percentage reduction in hospitalization rates for immunisation programs with and without relaxation of uptake and supply constraints compared to no vaccination, 2021–2022.**

|  | 2021 | | 2022 | | 2021–2022 | |
|---|---|---|---|---|---|---|
|  | Incidence | % | Incidence | % | Incidence | % |
| No vaccine | 12,722 | ref | 14 901 | ref | 27,622 | ref |
| **Immunization program under constraints** | | | | | | |
| Strong supply constraint | 12,819 | 0.8% | 9,436 | -36.7% | 22,053 | -20.2% |
|  | [11993;13001] | [-6;2] | [8732;10163] | [-41;-32] | [21733;22053] | [-21;-18] |
| Weak supply constraint | 11,527 | -9.4% | 5,669 | -62.0% | 17,932 | -35.1% |
|  | [11126;12573] | [-13;-1] | [5195;8659] | [-65;-42] | [16321;18737] | [-41;-27] |
| Uptake constraint | 10,960 | -13.9% | 5,270 | -64.6% | 16,626 | -39.8% |
|  | [9812;11996] | [-23;-6] | [4932;7392] | [-67;-50] | [14744;17266] | [-47;-33] |
| **Immunization program with relaxed constraints** | | | | | | |
| Relaxed strong supply and uptake constraint | 11,630 | -8.6% | 3,181 | -78.7% | 14,811 | -46.4% |
|  | [11294;12444] | [-11;-2] | [1826;4297] | [-88;-71] | [13846;14894] | [-50;-44] |
| Weak supply and uptake constraint | 9,606 | -24.5% | 1,702 | -88.6% | 11,497 | -58.4% |
|  | [9240;11204] | [-27;-12] | [1590;2110] | [-89;-86] | [11053;11497] | [-60;-52] |

Based on reference scenario for disease characteristics, level of NPI response (second wave threshold), vaccine profile (protection against infections) and NPIs
maintained until end of 2022. Incidence is given per million population and the % of variation is calculated in reference to the no vaccine counterfactual. For each
scenario, both values for our reference scenario and range are provided. The vaccine efficacy in the reference case is assumed to be 70% ranging from 50% to 90%.

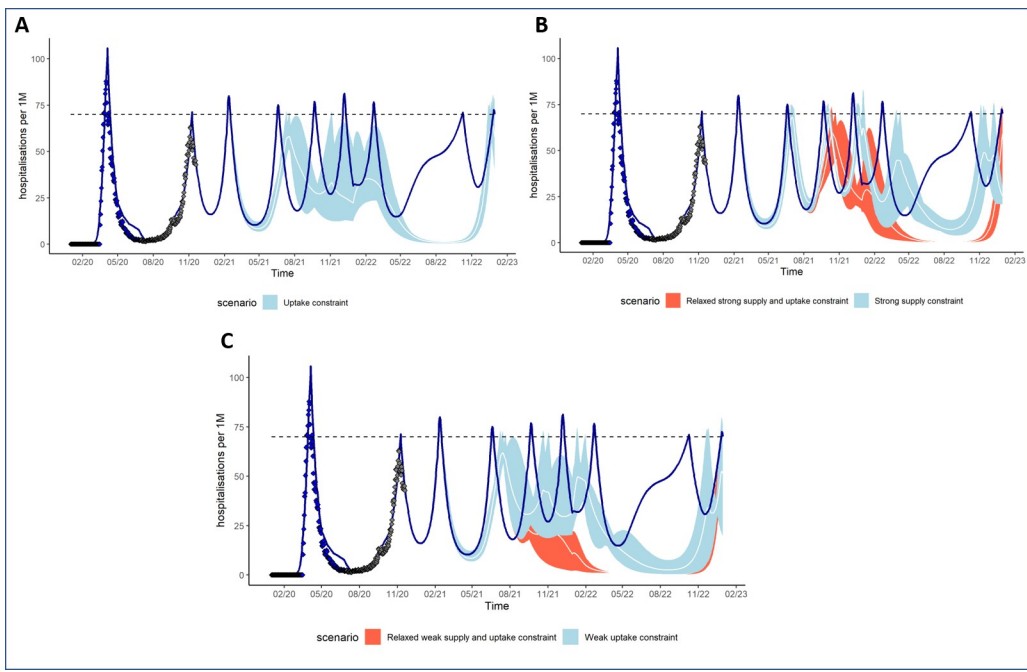

**Fig 3. Daily hospitalization incidence (rate per million population per day) with and without an immunisation program with varying uptake and supply constraints, 2020–2022.** Based on reference scenario for disease characteristics, NPI response (second wave threshold), vaccine profile (protection against infection) and NPIs maintained until end of 2022. **Panel A**–No vaccination (dark blue curve) and vaccination scenario under uptake constraint (light blue), **Panel B**–No vaccination (dark blue curve) and vaccination scenarios with strong supply constraint under two assumptions for relaxation of such constraints: the constraints not being eased during the second half of the year (light blue) or the constraints being eased during the second half of the years (orange), **Panel C**–No vaccination (dark blue curve) and vaccination scenarios with weak supply constraint under two assumptions for relaxation of such constraints: the constraints not being eased during the second half of the year (light blue) or the constraints being eased during the second half of the years (orange). The variation in impact due to the range of vaccine efficacy considered is shown as the area of the vaccine impact curves.

program scale-up (end of 2021). The corresponding evolution of daily hospitalization incidence is presented in Fig 3. In 2021, the median variation in hospitalizations is +0.8%, -9.4% and -13.9% if there is a strong supply constraint, a weak supply constraint, or an uptake constraint only, respectively. By the end of 2022, the variation in hospitalizations since the start of the program compared to no vaccine reaches 20.2%, 35.1%, and 39.8% for the same scenarios.

## Potential immunisation program impact if uptake and supply constraints are relaxed

In 2021, the level of additional health benefits associated to the relaxation of constraints remains moderate: the median reduction in hospitalization compared to no vaccination (Table 2) reaches 8.6% for the relaxed strong supply constraint scenario and 24.7% for the relaxed weak supply constraint scenarios. However, in 2022, the relaxed constraints scenarios are associated with further reductions in COVID-19 hospitalizations compared to no vaccination: respectively 78.7% for the relaxed strong supply constraint and 88.6% for the relaxed weak supply constraint. Similarly, to the no vaccine counterfactual scenario, the incidence remains significant at the end of 2022 for all scenarios with vaccination and even on the rise for most of them.

When assessing the incremental benefit of relaxing constraints, we observed an additional reduction in hospitalizations ranging respectively from 30 to 36% for the strong supply

**Table 3. Number of COVID-19 hospitalizations averted with and without relaxation of constraints in July 2021, 2021–2022.**

| | Without relaxation in July[1] | | With relaxation in July[2] | |
|---|---|---|---|---|
| | **Median** | **Range** | **Median** | **Range** |
| **Strong supply constraint** | | | | |
| Hospitalizations averted | 5,569 | [4,966;5,889] | 7,664 | [6,565;7,887] |
| Vaccinated subjects | 181,719 | [181,677;181,739] | 341,464 | [341,427;341,490] |
| Hosp averted/1,000 vaccinated subjects | 30.6 | [27.3;32.4] | 22.4 | [19.2;23.1] |
| Variation in incidence (%) | -20.2% | [-21; -18] | -34.6% | [-36;-30] |
| Days with NPIs | 393 | [387;498] | 269 | [207;285] |
| **Weak supply constraint** | | | | |
| Hospitalizations averted | 9,690 | [7,435;11,301] | 6,990 | [5,268;7,240] |
| Vaccinated subjects | 340,574 | [340,524;340,578] | 341,535 | [341,512;341,564] |
| Hosp averted/1000 vaccinated subjects | 28.5 | [21.8;33.2] | 20.5 | [15.4;21.2] |
| Variation in incidence (%) | -35.1% | [-41;-27] | -35.0% | [-39;-32] |
| Days with NPIs | 208 | [149;377] | 203 | [149;212] |

Based on reference scenario for disease characteristics, NPI response (second wave threshold), vaccine profile (protection against infection) and NPIs maintained until end of 2022.

1: Compared to the no vaccine counterfactual.

2: Compared to the corresponding constrained scenario. Hospitalizations averted: hospitalizations averted are per million population; Vaccinated subjects: people vaccinated per million population; Hosp averted/1,000 vaccinated subjects: Hospitalizations averted per 1,000 vaccinated people. For each scenario, both values for our reference scenario and range are provided. The vaccine efficacy in the reference case is assumed to be 70% ranging from 50% to 90%.

constraint scenario and 32 to 39% for the weak supply constraint scenarios (Table 3). This incremental benefit is associated with a decrease in the number of hospitalizations averted per 1,000 vaccinated people (respectively from 30.6 to 22.4 and from 28.5 to 20.5 for scenarios with strong or weak supply constraints before July). However, this apparent decrease in efficiency is compensated by a further decrease in the time with NPIs in place that is significant for the strong supply constraint scenarios (from 393 days without relaxation to 269 days with relaxation). The main benefit observed in scenarios with relaxation is their ability to prevent the needs for NPIs in 2022 (Fig 2).

Finally, in Fig 4, we present sensitivity analyses assessing the impact of uncertainty related to vaccine and disease characteristics, on the reduction of hospitalization incidence. In S1 File, we also present sensitivity analyses on the number of days with NPIs in 2022 (S6 Fig in S1 File). We used as a reference the relaxed strong supply and uptake constraints scenario and present results for the uptake constraint scenario in supplementary material (S7 Fig in S1 File). The variation in vaccine profile (from protecting against infection to only protecting against symptomatic disease) reduces the program impact from -46% in our reference case to -34%. This reduction has a larger impact than the one associated to a low efficacy (-44% if vaccine efficacy is 50%) and is also associated with a significant number of days with NPIs in 2022 (69 days).

Among uncertainties related to disease characteristics, duration of natural immunity and severity of reinfection have the largest impact. A median duration of natural immunity of two years is associated to a broader vaccination impact (-53%). On the opposite side, if reinfections are as severe as primary infections, the period with NPIs is predicted to exceed 100 days in 2022 (118 days) and the impact of vaccination drops to -27%. Therefore, if reinfections are as severe or even 50% less severe than initial infections, NPI activations would still be required in 2022 with the vaccination scenarios considered in our analysis.

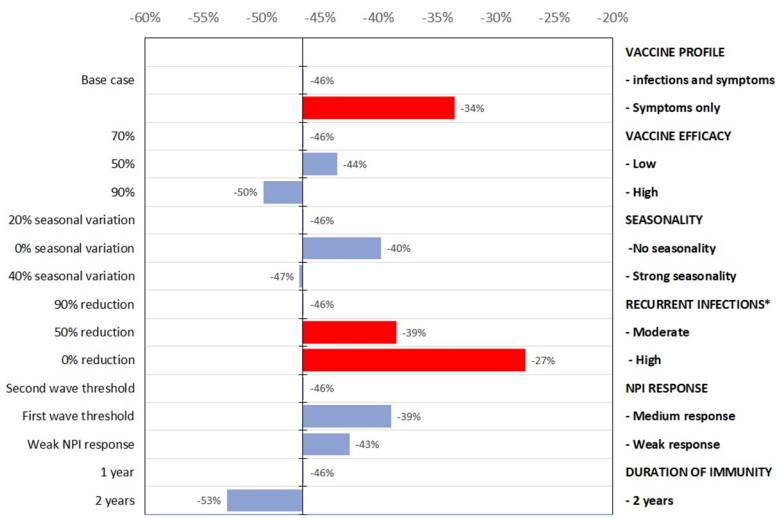

**Fig 4. Tornado diagram on the impact of a variation of vaccine and disease characteristics on the reduction in hospitalization in 2021–2022 associated to vaccination (relaxed strong supply and uptake constraints scenario).** All outcomes presented corresponds to univariate sensitivity analysis of the reference case for key disease and vaccine characteristics. The figure shows the change in number of hospitalizations (as %) for the 2021–2022 period compared to no vaccination counterfactual for different vaccine and disease characteristics. The red bars correspond to factor with the largest impact, figures next to the bars to the impact on COVID-19 hospitalizations compared to no vaccination and bars are oriented right or left depending of if the figure is smaller or higher than base case (first row).

## Discussion

We explored the short-term impact of an immunisation program with supply and uptake constraints changing over time. Our analysis confirmed that an adult immunisation program, even with limited supply and uptake, could significantly mitigate the health consequences of COVID-19, albeit not obviating the need for NPIs in the short term. This analysis is timely in the context of implementation and supply constraints faced by countries at the time of revised version writing (February 2021). It also helps contextualising the important of public messages on mitigation measures that are needed as the program is rolled out, and the challenges vaccine hesitancy represent to the success of local immunisation programs.

Our results are in accordance with previously published results of vaccine impact where coverage, the rate of vaccination, and efficacy play key roles in the government's ability to reduce social distancing measures [10, 12, 14, 17, 44–46]. Our analysis adds to previous literature in that it provides a detailed analysis of the likely implementation constraints and timing of benefits of a future immunisation program. In addition, we assessed the potential impact on the number of NPIs days, as an indicator of economic performance recovery. Assuming NPIs are maintained throughout until the end of 2022, a constrained immunisation program could result in 20% to 40% reduction in COVID-19 hospitalizations depending on the level of constraints compared to no vaccination over two years. Furthermore, NPIs may be avoided post-2021 depending on the extent of the constraints in place during roll-out. Relaxing both supply and uptake constraints towards mid-2021 increases the overall health impact and limits the risk of outbreaks once the program is completed and NPIs discontinued. The benefits of relaxing supply and uptake constraints start to be observed during the last quarter of 2021 and enable to a significant reduction in hospitalizations compared to the no vaccine counterfactual in 2021(exceeding 75%). However, to be successful, relaxation of such constraints requires achieving high vaccination coverage rates (from 68% to 89% of the adult population).

Our results on the incremental benefits associated to scenarios with relaxation of supply and uptake constraints indicate that lifting these constraints changes the value of the program over time as measured by the number of hospitalizations averted per 1000 people vaccinated (technical efficiency). However, this observation is the product of a trade-off: while it reduces the number of hospitalizations prevented per vaccination performed, it also reduces the need for NPIs and even prevents the need for them post-2021.

The insights on technical efficiency provided by our analysis are clearly only indicative. We do not account for the additional resources required to increase coverage nor do we capture the whole period for which vaccinations can impact COVID-19 outcomes. Given the scale and scope of societal impact of the COVID-19 pandemic and the urgency of the response, policy has been focused on alleviating health burden and curtailing societal and economic disruption. Quantifying the impact of an immunisation program focusing on hospitalizations averted and the need for NPI continuation in the short term allowed us to address both dimensions in an emergency response. While conceptually immunisation and social restriction responses could benefit from trade off analyses independently of the emergency of the response, our study emphasises that, after the initial public health response, expansions of immunisation programs will benefit from conventional assessments (such as cost-effectiveness assessments) of health, economic, and social welfare to optimise further policy responses [47].

With regards to the vaccine profiles explored, our primary analysis assumed vaccines available would protect against infection (and therefore against symptomatic disease) that can be seen as the most likely scenario. However, if any initially available vaccines only protect against symptomatic disease, although such vaccines would remain beneficial, we expect a reduced ability to prevent the need for NPI or outbreak occurrence after NPI discontinuation. We limited our analysis to vaccines offering a one-year protection. With such duration, the evolution of COVID-19 hospitalizations observed at the end of our period of analysis, at a time when most vaccine-conferred protection has waned, points to a need for revaccination to maintain disease control post-2022. Data on duration of protection, from immunity afforded by natural infection or vaccination, are starting to appear but it is early to know whether immunity will last longer than one year [48].

Among disease characteristics, severity of reinfection can also have a significant influence on the ability to control the disease not only in the long run but also in the next two years as considered here. To date, limited information is available on the severity of reinfection but our results indicate that, even with a broad vaccination program, NPIs would be still needed in 2022 to prevent a major COVID-19 outbreak if reinfection are as severe as primary infection. Finally, another aspect of uncertainty not addressed in our manuscript relates to changes in the virus to scape host immunity and/or improve its transmissibility [49]. Also, some mutations observed in the viruses suggest an ability to escape antibody immunity that could affect the success of vaccination programs in the short term [50] reinforcing the need for mitigation measures to be considered during 2021.

As with any modelling study, our work has limitations due to several simplifications and our results should be interpreted with caution. First, our characterisation of uncertainty remains limited. There are still many unknowns in the future evolution of COVID-19, public policy response, natural history and, importantly, the characteristics of vaccines to be approved. In this uncertain context, rather that assessing all possible scenarios including the most optimistic and pessimistic ones, we aimed at identifying the main drivers that could impact our conclusions. We proceeded to integrate and assess uncertainty in our analysis in several ways. We calibrated to several outcomes (hospitalizations, deaths, and cases reported), included uncertainty ranges linked to vaccine efficacy, and used scenario and sensitivity analyses. Secondly, even if we account for a reduced exposure to infection of vulnerable people and

a prioritisation in access to vaccination, our modelling framework do not account fully for the possible correlation between risk associated to COVID-19 and willingness to accept vaccination. Previous research on the profile of people reluctant to accept vaccination showed to include low-income people and people aged older than 75 years [19]. Therefore, our results may have overestimated the impact of vaccination in the presence of self-selection out of a program by individuals at higher risk of infection or complications. Vaccines have been introduced in France during January as modelled. However, there have been supply and implementation challenges in the program roll out. A successful roll out of vaccines in 2021 will allow for a significant impact over the 2021–2022 period. Our results showing a possible increase in COVID-19 activity later in 2022 results from the assumption that the immunisation program only lasts one year. The timing and magnitude of subsequent raises in activity in the long term will depend on other variable such as the duration of vaccinal and natural immunity, presence of routine immunisation programs or the severity of reinfection. Our results also might not apply directly to all settings as significant differences exist across the globe in terms of mitigation strategy and management of constraints. We did not explore the longer term as there is significant uncertainty on these different drivers or the impact of virus mutation. Finally, we do not formally assess the economic value of vaccines but present a proxy for programmatic (technical) efficiency. Our approach in this regard is deliberately conservative as it pertains to the use of vaccines in the short term to mitigate the pandemic effects.

This research has implications for vaccine research and development as well as policy. While a vaccine introduced in limited supply and uptake could positively impact the COVID-19 epidemic, additional doses or vaccines made available later in 2021 will help reduce the health burden further and prevent the need for NPIs post-2021. It is expected that uncertainty around vaccine characteristics will resolve as vaccination programs are implemented and data become available, yet there is a need to monitor the severity of reinfection in trials and post-regulatory approval commitments both in those people vaccinated and non-vaccinated, and for each vaccine in use. Experience gained on vaccines and their use in real conditions could improve vaccine acceptance in the population. Finally, immunisation is expected to play a central role in helping societies move on from this pandemic. Yet the efficiency of an immunisation program is likely to change as programs expand. A continuous assessment of the future value of vaccination within a comprehensive response to the pandemic will be needed to ensure optimal post-emergency use.

## Supporting information

**S1 File.**
(DOCX)

## Acknowledgments

We would like to recognize our colleagues from Sanofi Pasteur medical and R&D for insightful discussions about specific aspects of vaccination challenges. Their deep knowledge of industry, research and development, and modelling was invaluable when shaping our thinking.

## Author Contributions

**Conceptualization:** Laurent Coudeville, Ombeline Jollivet, Cedric Mahé, Sandra Chaves, Gabriela B. Gomez.

**Data curation:** Laurent Coudeville, Ombeline Jollivet, Gabriela B. Gomez.

**Formal analysis:** Laurent Coudeville, Ombeline Jollivet.

**Investigation:** Gabriela B. Gomez.

**Methodology:** Laurent Coudeville, Ombeline Jollivet, Sandra Chaves, Gabriela B. Gomez.

**Project administration:** Laurent Coudeville.

**Resources:** Gabriela B. Gomez.

**Software:** Laurent Coudeville, Ombeline Jollivet.

**Supervision:** Cedric Mahé.

**Validation:** Laurent Coudeville, Ombeline Jollivet, Cedric Mahé, Sandra Chaves, Gabriela B. Gomez.

**Visualization:** Laurent Coudeville, Ombeline Jollivet.

**Writing – original draft:** Laurent Coudeville, Ombeline Jollivet, Cedric Mahé, Sandra Chaves, Gabriela B. Gomez.

**Writing – review & editing:** Laurent Coudeville, Ombeline Jollivet, Cedric Mahé, Sandra Chaves, Gabriela B. Gomez.

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
