## [Decision Letter · Decision Letter 0]

8 Feb 2021

PONE-D-20-37719

Potential impact of introducing vaccines against COVID-19 under supply and uptake constraints in France: a modelling study

PLOS ONE

Dear Dr. coudeville,

Thank you for submitting your manuscript to PLOS ONE. After careful consideration, we feel that it has merit but does not fully meet PLOS ONE’s publication criteria as it currently stands. Therefore, we invite you to submit a revised version of the manuscript that addresses the points raised during the review process.

Your manuscript was reviewed by 2 experts in the field. Both identified many important problems in your submission and produced copious comments. Please review the attached comments and provide point-by-point responses.

We look forward to receiving your revised manuscript.

Kind regards,

Yury E Khudyakov, PhD

Academic Editor

PLOS ONE

Journal Requirements:

"This work was funded by Sanofi Pasteur."

We note that one or more of the authors have an affiliation to the commercial funders of this research study : Sanofi Pasteur..

3.1. Please provide an amended Funding Statement declaring this commercial affiliation, as well as a statement regarding the Role of Funders in your study. If the funding organization did not play a role in the study design, data collection and analysis, decision to publish, or preparation of the manuscript and only provided financial support in the form of authors' salaries and/or research materials, please review your statements relating to the author contributions, and ensure you have specifically and accurately indicated the role(s) that these authors had in your study. You can update author roles in the Author Contributions section of the online submission form.

3.2. Please also provide an updated Competing Interests Statement declaring this commercial affiliation along with any other relevant declarations relating to employment, consultancy, patents, products in development, or marketed products, etc.  

Reviewers' comments:

Reviewer's Responses to Questions

**Comments to the Author**

1. Is the manuscript technically sound, and do the data support the conclusions?

Reviewer #1: Yes

Reviewer #2: Yes

2. Has the statistical analysis been performed appropriately and rigorously? 

Reviewer #1: N/A

Reviewer #2: N/A

3. Have the authors made all data underlying the findings in their manuscript fully available?

Reviewer #1: Yes

Reviewer #2: Yes

4. Is the manuscript presented in an intelligible fashion and written in standard English?

Reviewer #1: Yes

Reviewer #2: Yes

5. Review Comments to the Author

Reviewer #1: The study is a statistic model of the vaccine effect on public health. the study adds to previous literature analysis on timing of benefits of future immunization program. The study is published when there are already preliminary data on the efficacy of the vaccine and therefore it is published a little too late (After the horses are already out of the barn). there is still benefit in publishing as the model predicts the morbidity in the next two years.

The study claim that the vaccine will need to be highly effective and to achieve high coverage to be able to obviate the need of non-pharmaceutical interventions and to control the pandemic. The data support the claim but there are some major comments:

Introduction:

1. Line 4- the numbers should be updated (for example there are more than 20 ongoing phase III trials and not 11).

2. Line 8- the number of vaccine candidates should be updated (there are at least 3 authorized vaccines already).

3. Line 23- please add reference to the study that was mentioned (modelling by the imperial college…).

4. Line 23- references 13 and 14 are actually a survey results about vaccine hesitancy and less about strategies to optimize immunization. Please add relevant reference.

5. Line 25- reference 15 is a study about the willingness of the population to be vaccinated. I couldn't find any support in the authors claim about supply constraint. Another reference would be more suitable.

Methods:

1. All the references should be checked. For example hospital admission reported by sante was specified as Ref 24 when it is actually Ref 25.

2. The definitions of uptake constrains and supply constrains are confusing and not intuitive. For example the phrase 'Relaxed uptake constraints' is confusing and not intuitively understood as high compliance. I suggest change 'uptake constraints' to more simple definition like 'vaccine compliance', and supply constrains to vaccine supply or quantity.

3. Line 43- please mention what period was taken.

4. Line 79- the association of other vaccines, such as influenza vaccine, have been proved to reduce COVID-19 infection(doi: 10.1080/21645515.2020.1852010). It can therefore be assumed that a dedicated vaccine will prevent disease and not only reduce symptoms. I suggest discuss it and take it into account when assuming vaccine efficacy.

5. Line 83- the authors mentioned efficacy of 50-90%. Previous studies mentioned 95% efficacy of the vaccine (for example: DOI: 10.1056/NEJMoa2034577). Such a large variation in the efficacy data of the vaccine may alter the results of the statistical model.

6. Line 99- the vaccine uptake was set as 60%. On what was the assessment based? There are published surveys on compliance that the author can rely on (for example: DOI: 10.1016/j.vaccine.2020.08.043).

7. Line 109 -the details are not accurate. There are already a few available vaccines. Assumption is not needed.

8. Line 130- the sentence is to complicate and long. Please simplify.

Results:

1. To my point of view, mean and C.I instead of median and range will be more appropriate.

2. Line 184- reduction in hospitalization range is not presented in table 3. Please clarify.

3. The effect of the vaccine compliance and supply was measured as hospitalization incidence. I think that number of infected patients is more appropriate. Please explain why did you choose that variable.

4. Previous study has demonstrated that hemoglobin A1C is a predictor of COVID‐19 Severity (doi: 10.1002/dmrr.3398 ). Has a connection between adherence to diabetes treatment and infectivity (R) and response to the vaccine been taken into account?

5. Currently, there is a growing evidence of side effects of the vaccine. Was it taken into account in calculating hospitalization rates and compliance rates?

Tables:

1. Table 2- please rechecked the numbers (hospitalization rates are higher under 'vaccine strong constraints' than under 'no vaccine' at all).

Reviewer #2: In this manuscript the authors present an analysis of the COVID-19 spreading dynamics in France after the initiation of vaccination drive. The incorporation of constraints from both the supply and demand side adds realism to the model. Some statements from the results section drew my attention however. They are : (a) that even with the uptake constraint i.e. maximal vaccine distribution, there would be NPIs being applied in 2022, (b) that with discontinuation of NPI at the end of 2021, there would be +0·8 percent, -9·4 percent and (presumably negative although this has not been indicated by authors) 13·9 percent variation in hospitalizations in 2021 with the three vaccine constraints relative to the no vaccine case, and (c) that the incidence of COVID-19 remains significant at the end of 2022 with all situations of vaccination and even on the rise for most of them. These statements are counter-intuitive and a bit pessimistic – most people are hoping for a return to normalcy by this fall or at least by the end of the year. I would like the authors to recognize and discuss this fact in detail.

The authors could consider several major and minor revisions. Some major revisions are given below:

1 There are certain modeling studies of vaccination dynamics which present a more optimistic view than the authors’ work. A key example is

Shayak B, Sharma MM and Mishra AK, “Impact of immediate and preferential relaxation of social and travel restrictions for vaccinated people on the spreading dynamics of COVID-19 : a model-based analysis,” available at

https://www.medrxiv.org/content/10.1101/2021.01.19.21250100v1

The references by Alvarez et. al. and Betti et. al. in the above manuscript are also optimistic. While I am aware that all these works were written after the authors’ manuscript, they must now be cited. It must be explained why the authors’ results differ from these analyses.

2 What is the role of the temporary immunity in generating the authors’ case trajectories ? In other words, if the vaccine immunity had been 2 years or 5 years, then what would the trajectories have been like ? Authors should perform simulations to demonstrate this.

3 What is the role of the bang-bang NPI control strategy in generating the authors’ bleak predictions ? Instead of this strategy, if a continuous NPI were applied or NPI gradually relaxed over time then what would the trajectories have looked like ? Simulations should be performed to analyse these questions.

4 Why there is an increase in hospitalization in 2021 with strong supply constraint relative to no vaccination ? Surely this is a surprising result. In continuation of the above, the authors should explore the solution space in much greater detail. They should clearly identify the scenarios where the outbreak ends in a reasonable time-frame instead of continuing on into 2023 and beyond. This should be used to motivate a discussion of effective vs ineffective vaccines and good vs bad policy decisions during the vaccination drive.

Apart from the above major concerns, there are several minor issues as well, as given below.

5 The population of France should be mentioned so that the supply constraints can be understood in terms of percentage population.

6 The introduction should be updated to reflect the current situation of vaccination drives. EUA granted to Pfizer, Moderna, Oxford/ Astra Zeneca, ICMR/ Bharat Biotech and Sputnik vaccines should be mentioned.

7 In Tables 1 and 2, the phrases “relaxed strong/weak supply and uptake constraint” is not clear to me. By relaxing a constraint one typically understands that the constraint does not exist any longer. However this does not seem to be what the authors imply.

8 Figure 4 is barely legible; moreover, it is difficult to understand the point attempted to be conveyed by the authors. The authors must improve the clarity of presentation here.

9 Factors like governmental support have not been considered.

10 Currently there are several viral strains such as B1.1.7 and B1.351 that might interfere with the effectiveness of vaccinations. It would be really impactful if the authors would mention about these variants as well.

11 Another important variable is the availability of required man-power which was not mentioned in the paper which might influence the supply and uptake constraints.

12 Altough it is a modelling study, but as the core topic was vaccination, some concepts on involvement of antibodies or other physiological concepts could have made the story more interesting. Not required though.

In summary, the Article as written presents a very strong claim without basing it on a sufficiently solid foundation. It also suffers from avoidable defects of presentation. Hence I recommend the authors to revise the manuscript along the lines indicated above and resubmit the revised version.

6. PLOS authors have the option to publish the peer review history of their article (what does this mean?). If published, this will include your full peer review and any attached files.

Reviewer #1: No

Reviewer #2: No

---

## [Author Response · Author response to Decision Letter 0]

23 Feb 2021

We would like to thank the editor and reviewers for their thorough review of our manuscript and insightful comments. In this revised version, we aimed at addressing each of the comments as indicated in our point-by-point response below. 

Reviewer #1: The study is a statistic model of the vaccine effect on public health. the study adds to previous literature analysis on timing of benefits of future immunization program. The study is published when there are already preliminary data on the efficacy of the vaccine and therefore it is published a little too late (After the horses are already out of the barn). there is still benefit in publishing as the model predicts the morbidity in the next two years.

We agree with the reviewer that other modelling studies have addressed the issue of impact of future immunization programs. These studies have explored the parameter space of vaccine efficacy. As the reviewer points out, efficacy trials have been published and we are now in the process of expanding vaccination coverage. An analysis placing the constraints (supply and uptake) we are having in program expansion is timely and needed to inform the current discourse. Currently, demand is outstripping supply, governments and health systems are facing challenges in implementation of what is arguably the biggest immunization program in history. In addition, uptake constraints (for example a reluctance to be vaccinated) could potentially pose serious challenges to programs in countries such as France. Therefore, our message can reinforce the need of programs to tackle these issues sooner rather than later. We tried to reinforce this message (of timeliness) for the benefit of readers in the first paragraph of the discussion.

The study claim that the vaccine will need to be highly effective and to achieve high coverage to be able to obviate the need of non-pharmaceutical interventions and to control the pandemic. The data support the claim but there are some major comments:

Introduction:

1. Line 4- the numbers should be updated (for example there are more than 20 ongoing phase III trials and not 11).

we updated these numbers – it is indeed a rapidly evolving field. If accepted, we will also ensure these number are up to date in the proofs. 

2. Line 8- the number of vaccine candidates should be updated (there are at least 3 authorized vaccines already).

we updated these numbers. If accepted, we will also ensure these number are up to date in the proofs. 

3. Line 23- please add reference to the study that was mentioned (modelling by the imperial college…).

Apologies to the reviewers and editor – we experienced a glitch when generating the reference list and the numbers do not correspond to the references meant. We have verified all references. 

4. Line 23- references 13 and 14 are actually a survey results about vaccine hesitancy and less about strategies to optimize immunization. Please add relevant reference.

same as above

5. Line 25- reference 15 is a study about the willingness of the population to be vaccinated. I couldn't find any support in the authors claim about supply constraint. Another reference would be more suitable.

same as above

Methods:

1. All the references should be checked. For example hospital admission reported by sante was specified as Ref 24 when it is actually Ref 25.

same as above

2. The definitions of uptake constraints and supply constraints are confusing and not intuitive. For example the phrase 'Relaxed uptake constraints' is confusing and not intuitively understood as high compliance. I suggest change 'uptake constraints' to more simple definition like 'vaccine compliance', and supply constrains to vaccine supply or quantity.

we have five scenarios of varying constraints. The terminology aims to reflect the programmatic evaluation literature and we want to highlight the limitation of programs as well as the dynamic aspect of the limitations. To clarify, we have added the following explanations to table 1. 

Uptake constraint reflects a limited coverage rate due to a limited willingness to be vaccinated in the population. 

Strong supply constraint reflects a limited amount of vaccines doses made available to the national program. In this case, the program is severely limited. 

Weak supply constraint reflects a limited amount of vaccines doses made available to the national program, in this case the program is moderately limited. 

Relaxed strong supply and uptake constraints. In this scenario, while the program is severely limited during the first half of the year, vaccine supply is increased during the second half of the year (higher production or new vaccines availability) and the public is more likely to be willing to vaccinate as the program has been in place for half a year. Therefore, we ‘eased’ the limitations of the program in the second semester. 

Relaxed weak supply and uptake constraints. In this scenario, while the program is moderately limited during the first half of the year, vaccine supply is increased during the second half of the year and the public is even more likely to be willing to vaccinate as the program has been in place for half a year. Therefore, we ‘eased’ the limitations of the program to achieve maximum coverage.

3. Line 43- please mention what period was taken.

we have added the duration of such immunity period – Median duration of one year. 

4. Line 79- the association of other vaccines, such as influenza vaccine, have been proved to reduce COVID-19 infection(doi: 10.1080/21645515.2020.1852010). It can therefore be assumed that a dedicated vaccine will prevent disease and not only reduce symptoms. I suggest discuss it and take it into account when assuming vaccine efficacy.

We agree with the reviewer that given experience gathered on other vaccines, the ability for vaccines to impact COVID-19 infections and hence reduce transmission is a likely scenario. We besides considered for our reference case a vaccine that prevents infection. This has however not been fully demonstrated to date even if some preliminary evidence exists (https://papers.ssrn.com/sol3/papers.cfm?abstract_id=3777268, https://doi.org/10.1101/2021.02.08.21251329)

Regarding the link between COVID-19 and influenza vaccination, even if the publication mentioned by the reviewer indicates positive impact of previous influenza vaccination, other studies have not found this association (https://academic.oup.com/occmed/article/70/9/665/6029444 and https://onlinelibrary.wiley.com/doi/10.1111/irv.12839). 

In this revised version, we indicate that a vaccine preventing infection is the most likely scenario and acknowledged remaining uncertainties. We however did not discuss the link between COVID-19 vaccination and previous influenza vaccination we considered is beyond the scope of our analysis.

5. Line 83- the authors mentioned efficacy of 50-90%. Previous studies mentioned 95% efficacy of the vaccine (for example: DOI: 10.1056/NEJMoa2034577). Such a large variation in the efficacy data of the vaccine may alter the results of the statistical model.

We have rephrased the paragraph on vaccine’s profiles modelled to reflect current knowledge available for already approved vaccines. The results are presented for a vaccine efficacy of 70% with shaded areas representing a variation from 50-90%. While efficacy in clinical trials with short follow up has been shown to be 95% for some of the vaccines, we wanted to provide a range that represents the mix of vaccine effectiveness (real life estimates) likely to be available, representing scenarios and not necessarily guessing actual estimates. Moreover, we are still uncovering results of vaccine effectiveness on reduction of infection (i.e., asymptomatic and symptomatic infections). We have added text to the figures, so that ranges of efficacy are clearly represented. 

6. Line 99- the vaccine uptake was set as 60%. On what was the assessment based? There are published surveys on compliance that the author can rely on (for example: DOI: 10.1016/j.vaccine.2020.08.043).

We based the 60% uptake on published surveys of willingness to vaccinate attitudes similar to the one suggested by the reviewer but specific to France. We have reworded the text to reflect the inclusion of this evidence. 

7. Line 109 -the details are not accurate. There are already a few available vaccines. Assumption is not needed.

we have reviewed the statements on vaccine availability in Europe to include further exploratory talks started mid-January and the authorizations to date. Our scenarios reflect supply constraints that are not only related to the authorization of vaccines but also the industrial capacity of companies to deliver at the scale needed. In fact, despite initial projections of vaccine dose availability, most countries in Europe and in North America are facing supply constraints due to production limitations (similar to the constraint scenarios assumptions). We added text to the methods to back up our assumptions. 

8. Line 130- the sentence is to complicate and long. Please simplify.

We have simplified the text to clarify what is present in the main text as opposed to the additional results.

Results:

1. To my point of view, mean and C.I instead of median and range will be more appropriate.

we present the values for our reference case, a vaccine efficacy of 70% and the range represents the variation in results of our range of efficacy considered. We have clarified this in each figure., table and the results. 

2. Line 184- reduction in hospitalization range is not presented in table 3. Please clarify.

The range mentioned line 184 (30 to 39) corresponds respectively to the lower and upper values of the confidence intervals corresponding to the two vaccination scenarios with constraint we considered in our manuscript. To clarify this point, we modified the sentence to indicate explicitly the two confidence intervals reported Table 3. 

3. The effect of the vaccine compliance and supply was measured as hospitalization incidence. I think that number of infected patients is more appropriate. Please explain why did you choose that variable.

we chose to present hospitalizations in the main text as it is a locally relevant indicator of the stress COVID-19 is putting on the health system which leads to government decisions on implementing mitigation measures. However, we acknowledge this is not the only relevant indicator. We present additional indicators such as number of days with NPIs in place in the main text, while cases and deaths are presented in the appendix. We have clarified this in the methods section and discussion. 

4. Previous study has demonstrated that hemoglobin A1C is a predictor of COVID‐19 Severity (doi: 10.1002/dmrr.3398 ). Has a connection between adherence to diabetes treatment and infectivity (R) and response to the vaccine been taken into account?

There may be an association between hemoglobin A1C and COVID-19 severity as pointed out by the reviewer and which is still explored by some researchers. However, the relevance of such predictor in the context of our analysis is accounted for by using broader risk categories driven by local literature on fatality ratio, hospitalization rates and social contact matrices. Moreover, our analysis did not aim to account for the effect of every individual risk factor, varying individual immune response and all the angles of constraints associated with rollout of vaccination programs for COVID-19. Our manuscript demonstrates scenarios driven by uncertainties and discuss the importance of considering such uncertainties when planning public health interventions.

5. Currently, there is a growing evidence of side effects of the vaccine. Was it taken into account in calculating hospitalization rates and compliance rates?

Currently, the hospitalization rates and uptake are based on the reported rates for France and the willingness to vaccinate on surveys, respectively. As both pieces of evidence are pre-vaccine, the hospitalization rates or uptake rates do not account for changes since the start of the program. However, the program started 3 weeks ago and there is no data available. We added this as a limitation to the discussion. 

Tables:

1. Table 2- please rechecked the numbers (hospitalization rates are higher under 'vaccine strong constraints' than under 'no vaccine' at all).

The result for the strong supply constraint scenario indeed indicates a potential slight increase in hospitalization in 2021 (range: -6 to +2%) but not for the 2021-2022 period (range : -18 to -21%) and not for all other scenarios. This rather counterintuitive result stems from the fact that the vaccination program indirectly impacts the level of NPI in our analysis which remains an important driver for the overall COVID-19 burden in 2021 (lower NPIs are associated with a larger number of hospitalizations). As it corresponds to a rather extreme scenario and is no longer observed when a more adequate time horizon for assessing vaccination benefits is considered (i.e. 2021-2022), this result should not be overstated. Nevertheless, it reinforces the message in our manuscript around the need for a combination of vaccination and NPI in 2021. 

Reviewer #2: In this manuscript the authors present an analysis of the COVID-19 spreading dynamics in France after the initiation of vaccination drive. The incorporation of constraints from both the supply and demand side adds realism to the model. Some statements from the results section drew my attention however. They are : (a) that even with the uptake constraint i.e. maximal vaccine distribution, there would be NPIs being applied in 2022, 

(b) that with discontinuation of NPI at the end of 2021, there would be +0·8 percent, -9·4 percent and (presumably negative although this has not been indicated by authors) 13·9 percent variation in hospitalizations in 2021 with the three vaccine constraints relative to the no vaccine case, 

and (c) that the incidence of COVID-19 remains significant at the end of 2022 with all situations of vaccination and even on the rise for most of them. 

These statements are counter-intuitive and a bit pessimistic – most people are hoping for a return to normalcy by this fall or at least by the end of the year. I would like the authors to recognize and discuss this fact in detail.

In this analysis, we focused in the short-term impact of a vaccination program taking place during 2021 with a specific focus on the role of supply and uptake constraints. Even if some our conclusions might appear pessimistic to the reviewer, our conclusions around the need to maintain NPIs alongside vaccination are actually consistent with several other recent pre-publications see e.g. https://doi.org/10.1101/2021.01.06.21249339 , https://doi.org/10.1101/2020.12.30.20248888 . In any case, our analysis can contribute to the debate around the need to tackle issues related to programmatic issues associated to vaccination programs

With regards to COVID-19 activity observed post 2021, our results notably reflect assumptions on duration of immunity and the fact that we did not specifically consider the implementation of routine vaccination in 2022 as it is beyond the scope of the analysis presented in our manuscript. 

We however acknowledge that there are clearly remaining issues with regards to the future evolution of COVID-19. In this revised version, we expanded the discussions notably the limitations of the study to hopefully clarify our conclusions and put them in regards with the current knowledge on COVID-19. 

We also thank the reviewer for identifying the missing minus sign before 13.9, this was corrected in main text (already accurate in table 2).

The authors could consider several major and minor revisions. Some major revisions are given below:

1 There are certain modeling studies of vaccination dynamics which present a more optimistic view than the authors’ work. A key example is

Shayak B, Sharma MM and Mishra AK, “Impact of immediate and preferential relaxation of social and travel restrictions for vaccinated people on the spreading dynamics of COVID-19 : a model-based analysis,” available at

https://www.medrxiv.org/content/10.1101/2021.01.19.21250100v1

The references by Alvarez et. al. and Betti et. al. in the above manuscript are also optimistic. While I am aware that all these works were written after the authors’ manuscript, they must now be cited. It must be explained why the authors’ results differ from these analyses.

thank you for pointing these papers out. We have referenced and added a commentary in the discussion to place our results in the context of these other studies. 

2 What is the role of the temporary immunity in generating the authors’ case trajectories ? In other words, if the vaccine immunity had been 2 years or 5 years, then what would the trajectories have been like ? Authors should perform simulations to demonstrate this.

We made the choice in this manuscript to focus on programmatic constraints without necessarily exploring in detail the duration of immunity, especially as our period of analysis is relatively short (2 years including one year of vaccination). This is the topic of other papers in the field (see e.g. https://doi.org/10.1126/science.abd7343) and in a context where the short-term evolution of COVID-19 remains mainly driven by the primary infections and level of NPIs this does not necessarily significantly impact our conclusions even if a longer duration of immunity improved expected vaccination benefits.

Below, we illustrate the evolution in the absence of vaccination for a duration of immunity varying from 1 to 5 years. For a 2 years duration of naturally immunity, it is still expected to have a significant COVID-19 circulation in 2022. 

In the revised version, we updated Figure 4 to include a scenario with a media duration of immunity of 2 years and also added discussion on this point. 

3 What is the role of the bang-bang NPI control strategy in generating the authors’ bleak predictions ? Instead of this strategy, if a continuous NPI were applied or NPI gradually relaxed over time then what would the trajectories have looked like ? Simulations should be performed to analyse these questions.

We considered in our analysis two types of NPIs: threshold-based NPIs i.e. more stringent measures implemented when incidence exceeds a predefined threshold but also risk-based NPIs i.e. reduced exposure to infection for vulnerable people compared to the rest of the population. Contrary to the first one, this second one is continuous throughout the roll-out of the vaccination program. 

The threshold-based NPIs, that can be seen as a stop-and-go approach however account for a gradual relaxation of measures over time. Time-varying measures are also in fact consistent with actual measures implemented in a number of countries including France. Fully constant NPIs will not necessarily represent reality (even if authorities do not enforce any change in their policy a more active virus circulation can impact individual behavior). Besides, one of the main expected outcomes of vaccination is the ability to stop NPIs and the impact of vaccination on NPIs is part of the analysis we performed. 

We however agree with the reviewer that more constant NPIs could have been considered in our analysis (e.g. longer period for relaxation of NPIs after an outbreak) but this won’t have changed our main conclusion. In this revised, we explicitly mention as a limitation that we did not account for all possible evolutions of NPIs over time.

4 Why there is an increase in hospitalization in 2021 with strong supply constraint relative to no vaccination ? Surely this is a surprising result. 

(similar response to reviewer 1): The result for the strong supply constraint scenario indeed indicates a potential slight increase in hospitalization in 2021 (range: -6 to +2%) but not for the 2021-2022 period (range : -18 to -21%) and not for all other scenarios. This rather counterintuitive result stems from the fact that the vaccination program indirectly impacts the level of NPI in our analysis which remains an important driver for the overall COVID-19 burden in 2021 (lower NPIs are associated with a larger number of hospitalizations). As it corresponds to a rather extreme scenario and is no longer observed when a more adequate time horizon for assessing vaccination benefits is considered (i.e. 2021-2022), this result should not be overstated. Nevertheless, it reinforces the message in our manuscript around the need for a combination of vaccination and NPI in 2021. 

In continuation of the above, the authors should explore the solution space in much greater detail. They should clearly identify the scenarios where the outbreak ends in a reasonable time-frame instead of continuing on into 2023 and beyond. This should be used to motivate a discussion of effective vs ineffective vaccines and good vs bad policy decisions during the vaccination drive.

We focused our scenarios in the next 2 years due to the substantial uncertainties of a long-term modelling. The increase seen post-2022 is driven by the assumption that both natural and vaccine induced immunity last on average 1 year. This assumption comes from (population based and challenge) studies of human coronaviruses in the 80s and later. There are yet limited data to support long lasting assumptions (i.e., longer than 1 year). This is particularly relevant as we consider changes in the human antibody immune response to variant viruses already circulating. We have added this discussion to the manuscript more explicitly and indicated how the various constraint scenarios can affect the impact of vaccination in curtailing the pandemic effect, addressing reviewer’s concern.

Apart from the above major concerns, there are several minor issues as well, as given below.

5 The population of France should be mentioned so that the supply constraints can be understood in terms of percentage population.

the percentage coverage of the total population for all scenarios are presented in table 1, as well as for the subgroups. We added a reference to the total population for clarity as suggested. 

6 The introduction should be updated to reflect the current situation of vaccination drives. EUA granted to Pfizer, Moderna, Oxford/ Astra Zeneca, ICMR/ Bharat Biotech and Sputnik vaccines should be mentioned.

We have updated the text to reflect advances in the pipeline, the authorizations as well as implementation to date. 

7 In Tables 1 and 2, the phrases “relaxed strong/weak supply and uptake constraint” is not clear to me. By relaxing a constraint one typically understands that the constraint does not exist any longer. However this does not seem to be what the authors imply.

(similar response to reviewer 1): we have five scenarios of varying constraints. The terminology aims to reflect the programmatic evaluation literature and we want to highlight the limitation of programs as well as the dynamic aspect of the limitations. To clarify, we have added the following explanations to table 1. 

Uptake constraint reflects a limited coverage rate due to a limited willingness to be vaccinated in the population. 

Strong supply constraint reflects a limited amount of vaccines doses made available to the national program. In this case, the program is severely limited. 

Weak supply constraint reflects a limited amount of vaccines doses made available to the national program, in this case the program is moderately limited. 

Relaxed strong supply and uptake constraints. In this scenario, while the program is severely limited during the first half of the year, vaccine supply is increased during the second half of the year (higher production or new vaccines availability) and the public is more likely to be willing to vaccinate as the program has been in place for half a year. Therefore, we ‘eased’ the limitations of the program in the second semester. 

Relaxed weak supply and uptake constraints. In this scenario, while the program is moderately limited during the first half of the year, vaccine supply is increased during the second half of the year and the public is even more likely to be willing to vaccinate as the program has been in place for half a year. Therefore, we ‘eased’ the limitations of the program to achieve maximum coverage.

8 Figure 4 is barely legible; moreover, it is difficult to understand the point attempted to be conveyed by the authors. The authors must improve the clarity of presentation here.

As there is still a large number of uncertainties on COVID-19 evolution, we used a tornado diagram approach for assessing separately the potential impact of important uncertainty factors that is presented Figure 4. 

To facilitate the readability of this figure, in the revised version, we only kept in the main text the panel A (univariate sensitivity analysis on reduction over the 2021-2022 period) and added panel B to supplementary material). In response to the comment on duration of immunity, we also added a sensitivity analysis on this topic. 

9 Factors like governmental support have not been considered.

Governments have supported vaccine development by providing push and pull incentives – these include financing of R&D and production at risk as well as the streamlining of regulatory processes. We touch upon these in the introduction. Once the vaccine has been developed, in France, the design and implementation of the immunization program is government driven and thus, it is intrinsically supported by the national public health and health systems. We added some background around this narrative to the program description. 

10 Currently there are several viral strains such as B1.1.7 and B1.351 that might interfere with the effectiveness of vaccinations. It would be really impactful if the authors would mention about these variants as well.

We agree with the reviewer that the emergence of new variants with reported increased transmissibility and/or severity and threats for vaccine-induced immunity is a critical factor for the future evolution of COVID-19. However, the evidence gathered around these new variants is very recent and a complete analysis of their consequences is beyond the scope of our manuscript.

In this revised version, we added the discussion a paragraph of the potential consequences of these new variants that actually reinforces our conclusion around the need for NPIs alongside vaccination 

11 Another important variable is the availability of required man-power which was not mentioned in the paper which might influence the supply and uptake constraints.

The programs we modelled included assumptions on the ability of the health system to scale up, especially in the first half of 2021. The rate of vaccination is influenced by the availability of human resources, building space and consumables among others. We added this explanation to the methods.

12 Although it is a modelling study, but as the core topic was vaccination, some concepts on involvement of antibodies or other physiological concepts could have made the story more interesting. Not required though.

Thank you for your comment. Although we agree with the reviewer, it is difficult to extrapolate and expand on aspects of immunity in a manuscript that used modeling to discuss uncertainties related to the evolution of the pandemic in a broad population sense. However, we have added to the discussion on the aspects of vaccine response to give more depth to the paper.

In summary, the Article as written presents a very strong claim without basing it on a sufficiently solid foundation. 

our conclusion is that a limited immunization program may not be sufficient in the short term to avoid further spread of SARS-CoV-2. If the constraints of such an immunization program are addressed in the second half of the year, the impact increases and more control over the epidemic is realized. With this conclusion we aim to highlight the importance of considering vaccination impact in realistic contexts which includes implementation and supply constraints and vaccine hesitancy in the short term. We hope with the additional text included in the methods, results and discussion we have addressed this concern.

It also suffers from avoidable defects of presentation. 

we have addressed these defects as well as issues with the reference list. Thank you

Hence I recommend the authors to revise the manuscript along the lines indicated above and resubmit the revised version

To the best of our ability we have addressed both reviewers’ comments. Thank you

---

## [Decision Letter · Decision Letter 1]

22 Mar 2021

PONE-D-20-37719R1

Potential impact of introducing vaccines against COVID-19 under supply and uptake constraints in France: a modelling study

PLOS ONE

Dear Dr. coudeville,

Thank you for submitting your manuscript to PLOS ONE. After careful consideration, we feel that it has merit but does not fully meet PLOS ONE’s publication criteria as it currently stands. Therefore, we invite you to submit a revised version of the manuscript that addresses the points raised during the review process.

Your revised manuscript was reviewed by 2 experts in the field who reviewed the original version. Although one reviewer was completely satisfied with your modification of the manuscript, the other still identified many important issues and produced a very strong recommendation. Please consider carefully the attached comments and provide point-by-point responses

We look forward to receiving your revised manuscript.

Kind regards,

Yury E Khudyakov, PhD

Academic Editor

PLOS ONE

Reviewers' comments:

Reviewer's Responses to Questions

**Comments to the Author**

1. If the authors have adequately addressed your comments raised in a previous round of review and you feel that this manuscript is now acceptable for publication, you may indicate that here to bypass the “Comments to the Author” section, enter your conflict of interest statement in the “Confidential to Editor” section, and submit your "Accept" recommendation.

Reviewer #1: All comments have been addressed

Reviewer #2: (No Response)

2. Is the manuscript technically sound, and do the data support the conclusions?

Reviewer #1: Yes

Reviewer #2: Partly

3. Has the statistical analysis been performed appropriately and rigorously? 

Reviewer #1: Yes

Reviewer #2: N/A

4. Have the authors made all data underlying the findings in their manuscript fully available?

Reviewer #1: Yes

Reviewer #2: Yes

5. Is the manuscript presented in an intelligible fashion and written in standard English?

Reviewer #1: Yes

Reviewer #2: Yes

6. Review Comments to the Author

Reviewer #1: Appologies for the delay in reviewing the submitted article. after reading the author's comments, all required questions have been answered and all responses met formatting specifications.

Reviewer #2: Manuscript ID : PONE-D20-37719

Revision stage : First revision

Title : Potential impact of introducing vaccines against COVID-19 under supply and uptake constraints in France : a modeling study

Authors : Coudeville L et. al.

I thank the authors for revising their manuscript in response to the reviewers’ feedbacks. However, upon detailed reviewing of the revised manuscript, I am still not convinced about the utility of authors’ conclusions in a practical scenario.

My specific concern had been regarding the authors’ prediction that even with vaccination drive, COVID-19 would be a significant presence as late as 2023. To this end, I had asked the authors to consider various situations such as longer-lasting immunity of the vaccines and different pattern of NPIs so that we could get a much clear idea of the circumstances where COVID-19 continued for a long time and those where the disease got eliminated. I appreciate the fact that this analysis has indeed been performed by the authors. However, they have found high caseload of COVID-19 in 2023 in all situations, even with the most effective vaccine, the most relaxed constraint, and the longest immunity.

I am afraid that an epidemic model which can only produce solutions of one class is fundamentally limited or flawed. A versatile model must be able to show different kinds of solutions for different parameter values. One example is the model in

Shayak B, Sharma MM and Mishra AK, “COVID-19 spreading dynamics in an age-structured population with selective relaxation of restrictions for vaccinated individuals : a mathematical modeling study” (2021) available at https://www.medrxiv.org/content/10.1101/2021.02.22.21252241v1

In this Article it will be seen that the disease is getting contained above a threshold vaccine efficacy and perpetuated below that efficacy. I would like to remind the authors that full or near-total containment of COVID-19 without vaccination has already been achieved in New Zealand, Australia and Taiwan, so vaccine-aided elimination of the disease is not entirely a utopian concept. Hence, a model which cannot exhibit this solution in any case is not a very accurate description of reality. Just to be clear, I am not predicting that COVID-19 will definitely get contained by year-end. It is indeed possible that it will continue into 2023 like the authors predict. However, in my perspective a good mathematical model must be able to generate both classes of solutions and not just one of them. Thereafter, one can have a discussion regarding the situations which lead to the two different outcomes.

My confidence in the authors’ model is further dented by their prediction that vaccination under strong constraints would lead to greater number of hospitalizations in 2021 than no vaccine at all. I would have believed that even if one thousand people were to be vaccinated all over France then overall there would be several hundred less hospitalizations than if the vaccine were not administered.

In conclusion, the authors have apparently used an imperfect mathematical model to predict a dystopian scenario with COVID-19 continuing at full force into 2023. I am not confident of approving for publication such a pessimistic prediction that did not consider other possibilities in the trajectory. Hence, I must recommend that the manuscript be rejected. Perhaps, a suggestion would be to work on models with greater predictive power having more realistic applications.

7. PLOS authors have the option to publish the peer review history of their article (what does this mean?). If published, this will include your full peer review and any attached files.

Reviewer #1: No

Reviewer #2: No

---

## [Author Response · Author response to Decision Letter 1]

29 Mar 2021

Reviewer #1:

 Apologies for the delay in reviewing the submitted article. after reading the author's comments, all required questions have been answered and all responses met formatting specifications.

We are pleased to read that the reviewer is satisfied by the responses we gave to his previous comments 

Reviewer #2: 

I thank the authors for revising their manuscript in response to the reviewers’ feedbacks. However, upon detailed reviewing of the revised manuscript, I am still not convinced about the utility of authors’ conclusions in a practical scenario.

My specific concern had been regarding the authors’ prediction that even with vaccination drive, COVID-19 would be a significant presence as late as 2023. To this end, I had asked the authors to consider various situations such as longer-lasting immunity of the vaccines and different pattern of NPIs so that we could get a much clear idea of the circumstances where COVID-19 continued for a long time and those where the disease got eliminated. I appreciate the fact that this analysis has indeed been performed by the authors. However, they have found high caseload of COVID-19 in 2023 in all situations, even with the most effective vaccine, the most relaxed constraint, and the longest immunity.

I am afraid that an epidemic model which can only produce solutions of one class is fundamentally limited or flawed. A versatile model must be able to show different kinds of solutions for different parameter values. One example is the model in

Shayak B, Sharma MM and Mishra AK, “COVID-19 spreading dynamics in an age-structured population with selective relaxation of restrictions for vaccinated individuals : a mathematical modeling study” (2021) available at https://www.medrxiv.org/content/10.1101/2021.02.22.21252241v1

In this Article it will be seen that the disease is getting contained above a threshold vaccine efficacy and perpetuated below that efficacy. I would like to remind the authors that full or near-total containment of COVID-19 without vaccination has already been achieved in New Zealand, Australia and Taiwan, so vaccine-aided elimination of the disease is not entirely a utopian concept. Hence, a model which cannot exhibit this solution in any case is not a very accurate description of reality. Just to be clear, I am not predicting that COVID-19 will definitely get contained by year-end. It is indeed possible that it will continue into 2023 like the authors predict. However, in my perspective a good mathematical model must be able to generate both classes of solutions and not just one of them. Thereafter, one can have a discussion regarding the situations which lead to the two different outcomes.

Models exploring all parameter space to identify the ideal optimal combination of vaccine efficacy and coverage have been published (as referenced in our introduction). This type of approaches, that can be qualified as normative, are very useful in an initial phase of thinking through how to introduce technologies. They have been invaluable to define prioritisation policies ensuring societies aim to minimise deaths while maximising transmission reduction later in the pandemic. 

Currently, we are introducing vaccines in real programs and implementation of these programs is undeniably constrained. Our intent is to highlight the impact of the implementation constraints faced in France (as an example for EU countries). Therefore, rather than assessing all possible scenarios and identify optimal ones we selected a positive rather than normative approach which consists in identifying plausible scenarios given realistic situation and expected constraints. We see our approach as complementary to existing publications. 

The comparison performed by the reviewer between our results for France and the situation in Australia, New Zealand and Taiwan does not seem to us as the most relevant. First, the geographic and insular situation of these countries is very different from the one of France. Secondly, the near containment obtained under strict NPI measures cannot be really compared with the situation that could be observed later in France without such measures.

Failure to account for constraints during the priority setting process/implementation planning in mathematical modelling can result in unfeasible health interventions being recommended, theoretically optimal but practically unreasonable expectations of impact and, ultimately, in evidence being disregarded by decision-makers and public wariness. The need for a diversity of approaches in modelling analyses including accounting for constraints has been evident for years in resource limited settings, especially when introducing large intervention programs – such as a global vaccination campaign (see e.g. https://doi.org/10.1371/journal.pmed.1002240) The measurement, inclusion and analysis of impact of constraints in mathematical modelling is a change in paradigm in model-based policy recommendations as well described by Bozzani et al. in recent publication, quote, “Common objectives of model-based analyses incorporating constraints are to assess real-world feasibility or allocate resources efficiently” (see e.g. https://doi.org/10.1016/j.epidem.2021.100450)

We believe this perspective is needed in the literature, as we are facing a complicated set of constraints to scale up vaccination. 

To clarify this point in the revised version of the manuscript, we specifically indicate that our objective in this manuscript was not to identify optimal conditions for a vaccination program to be successful (line 23), that we did specifically assessed all possible scenarios including the most optimistic and pessimistic ones (line 286) and that our results are not directly applicable to all settings (line 304).

It is also noteworthy to mention that the situation we described actually reflects the current situation in France and in other countries (e.g. Germany, Belgium, Spain) where despite the initiation of the vaccination program, supply and uptake constraints are still very much present and has not prevented the need for lockdown measures which are currently being extended in most European countries. 

The conclusion in our manuscript is similar to that reported for other settings described in recent publications (e.g. Makhoul et al. [2021] that notably state “:Despite 95% efficacy, actual vaccine impact could be meager in such countries if vaccine scale-up is slow, acceptance is poor, or restrictions are eased prematurely.” https://doi.org/10.3390/vaccines9030223

My confidence in the authors’ model is further dented by their prediction that vaccination under strong constraints would lead to greater number of hospitalizations in 2021 than no vaccine at all. I would have believed that even if one thousand people were to be vaccinated all over France then overall there would be several hundred less hospitalizations than if the vaccine were not administered.

The correct way to interpret the result mentioned by the reviewer, is actually more a neutral impact of vaccination (range: -6 to +2%). It certainly does not mean that vaccination has no impact.

It rather indicates that the implementation of a vaccination program also impacts the NPIs in place (we report a reduction in NPI of 30 days in 2021 for the strong supply constraint scenario, see Figure 2). The overall impact is, therefore, a combination of vaccination effect and changes in NPIs measures. If the vaccination coverage is too low, changes in NPIs might have a stronger impact on hospitalizations than the vaccination itself. As it corresponds to a rather extreme scenario, is no longer observed when a more appropriate time horizon is considered (i.e. 2021-2022), the importance of this result is not to be overstated but it would be incorrect to dismiss it as a ‘wrong’ result – the peril of early relaxation of NPIs with suboptimal vaccination coverage is real and has been highlighted by others, notably Moore et al ‘For all vaccination scenarios we investigated, our predictions highlight the risks associated with early or rapid relaxation of NPIs. ( https://www.thelancet.com/journals/laninf/article/PIIS1473-3099(21)00143-2/fulltext)

Therefore, It does not seem for us that this result invalidates the quality of the model. Moreover, as we observed earlier, the impact of low vaccine uptake and supply constraints have led EU nations to extend their mitigations measures and/or initiate lockdowns after short period of relaxation – similar to what we discussed in our paper. 

In conclusion, the authors have apparently used an imperfect mathematical model to predict a dystopian scenario with COVID-19 continuing at full force into 2023. I am not confident of approving for publication such a pessimistic prediction that did not consider other possibilities in the trajectory. Hence, I must recommend that the manuscript be rejected. Perhaps, a suggestion would be to work on models with greater predictive power having more realistic applications.

We agree with the reviewer on the imperfectness of our model which is in fact a common trait to all models. However, we disagree on two aspects of the statement made by the reviewer:

 • Our scenarios are not dystopian. Implementation constraints are being reported daily and the risk of suboptimal vaccination in combination with early relaxation of control measures has been highlighted by several modelling groups to date in other settings. 

 • We do not predict ‘COVID-19 will continue in full force in 2023’. Our results indicate in most scenarios a limited need for NPIs in 2022 (e.g. 0-11 NPI days for most vaccination scenarios in 2022 except the “strong supply constraint” scenario) and this also applies to 2023. This cannot be compared to the 2020-2021 when significant outbreaks are observed despite the implementation of significant NPI measures. We still do not know the duration of natural or vaccine-afforded immunity. Based on experience with human coronaviruses, immunity from natural infection may last 1-2 years. We also do not know the level of severity expected in those re-infected. Therefore, we focused our model and discussion to a mid-term situation.

Each model is designed to fit the ability to answer a precise question and to the available information to fuel it and to validate it. We built a model aiming to be able to be sufficiently precise to be able to compare conditional scenario – ‘what if’ scenario – of public policies, with the most up-to-date knowledge of realistic constraints. Scenario planning is useful in considering alternative futures – so that policy makers can assess the opportunity costs of inaction. To consider scenario planning as prediction modelling has been a recurrent misinterpretation during this pandemic – leading to misplaced suspicion of modelling results. 

Finally, we are confident that our manuscript constitutes a positive addition to the scientific debate on the topic. It has, of course, to be put in perspective with other publications, which we hope we did.

---

## [Editor Report · Decision Letter 2]

14 Apr 2021

Potential impact of introducing vaccines against COVID-19 under supply and uptake constraints in France: a modelling study

PONE-D-20-37719R2

Dear Dr. coudeville,

We’re pleased to inform you that your manuscript has been judged scientifically suitable for publication and will be formally accepted for publication once it meets all outstanding technical requirements.

Kind regards,

Yury E Khudyakov, PhD

Academic Editor

PLOS ONE
---

## [Editor Report · Acceptance letter]

19 Apr 2021

PONE-D-20-37719R2 

Potential impact of introducing vaccines against COVID-19 under supply and uptake constraints in France: a modelling study 

Dear Dr. Coudeville:

I'm pleased to inform you that your manuscript has been deemed suitable for publication in PLOS ONE. Congratulations! Your manuscript is now with our production department. 

Kind regards, 

on behalf of

Dr. Yury E Khudyakov 

Academic Editor

PLOS ONE